# Soil Available Nitrogen and Yield Effect under Different Combinations of Urease/Nitrate Inhibitor in Wheat/Maize Rotation System

Xiumin Cui [1], Jingquan Wang [1], Jiahui Wang [1], Yun Li [1], Yanhong Lou [1], Yuping Zhuge [1] and Yuxiu Dong [2,*]

[1] College of Resources and Environment, Shandong Agricultural University, Tai'an 271018, China
[2] Key Laboratory of Crop Biology, Shandong Agricultural University, Tai'an 271018, China
* Correspondence: jingtianweidi2007@126.com; Tel.: +86-139-0548-8927

**Abstract:** In a wheat/maize rotation system, nitrogen (N) accounts for a large proportion of basal fertilizer, but soil N loss and the resulting environmental risk simultaneously exist worldwide. This study applied different urease/nitrification inhibitors together with basal fertilizers and investigated their effects on soil N level and grain yield. Six N stabilizing combinations consisted of two urease inhibitors (HQ and NBPT) and three nitrification inhibitors (DCD, DMPP, and Nitrapyrin). The treatments supplied with urease/nitrification inhibitors reduced, to some degree, the conversion rate of $NH_4^+$ into $NO_3^-$, and kept $NH_4^+$ content higher in surface soils for a longer time. Compared to CK, A1 treatment supplied with 1.5% HQ + 4% DCD well-maintained the levels of soil alkali-hydrolyzable N and $NH_4^+$. For example, alkali-hydrolyzable N and $NH_4^+$ contents at 0–20 cm soil layer under A1 were increased by 8.59–41.6% and 8.15–14.5% more than CK, respectively. Based on the entire growth period of wheat and maize rotation, urease/nitrification inhibitors improved soil available N in surface soils but did not prevent $NH_4^+$ and $NO_3^-$ leaching, especially in the intensive rainfall season. The combinations of HQ and DCD or Nitrapyrin significantly enhanced crop yield. Specifically, crop yields under A1 and A3 (1.5% HQ + 0.25% Nitrapyrin) were 16.3% and 14.3% higher than CK, respectively. The N stabilizing combinations also promoted N intake and transport at every growth stage. The maximum N accumulation was increased by 27% under A1, when compared to CK. The treatments supplied with urease/nitrification inhibitors also achieved higher apparent N recovery efficiency, N agronomic efficiency, and N partial factor productivity. Consequently, the combinations of urease/nitrification inhibitors could improve N availability at 0–40 cm soil layer, which in turn improved N use efficiency of wheat and maize. The results suggested that the two urease/nitrification inhibitor combinations, 1.5% HQ + 4% DCD (A1) and 1.5% HQ + 0.25% Nitrapyrin (A3), were optimal N stabilizing agents and worthy of further study.

**Keywords:** available nitrogen; nitrogen leaching; nitrogen use efficiency; increase production

## 1. Introduction

Nitrogen (N) element is crucial for crop yield, but N deficiency is widely existing in agricultural practices, thus supplying nitrogen fertilizer is one of the most effective measures to increase grain yield. However, the phenomenon of excessively applying fertilizer commonly exists due to the low educational level of farmers or employees. As the law of diminishing returns has shown, the higher harvest does not occur correspondingly as expected, while simultaneous deficiency of available N is not yet the main limiting factor for crop yield [1]. It has been reported that the N fertilizer application rate of China is 75% more than the world's average level, while N use efficiency is only 30–40% of the world's average [2,3]. Due to the surplus input and lower utilization coefficient, a large amount of N fertilizer is lost consequently through $NH_3$ volatilization, $N_2O$ emissions, and nitrate leaching loss, which is not only a resource waste but also causes environmental pollution [2–5].

Urea [$(NH_2)_2CO$] is widely used as the N source in both simple and compound fertilizer. After going into soils, urea molecule is easily hydrolyzed into $(NH_4)_2CO_3$ by urease and then releases $NH_3$, which is one of the main ways of N loss, and also the main source of $NH_3$ in airborne aerosols and haze [6,7]. On the other hand, the superfluous $NH_4^+$ in soil solutions and $NH_4^+$ absorbed on soil colloid are prone to nitrificate in a well-ventilated environment, and can leach along with natural precipitation and artificial irrigation. While in a poor-ventilated environment, $NO_3^-$ could be easily converted into $N_2O$ or $N_2$ by denitrifying bacteria and then lose from soils, thus also lowering of the N use efficiency [8,9].

Huang-Huai-Hai region (HHHR) is a significant grain base of China, where N fertilizer accounts for 70–90% of the total basal fertilizers in the maize/wheat rotation system, and urea is one of the main N fertilizer sources [4,6]. Studies have found that it takes 7–10 days, 4–5 days, and 2 days to convert urea into $N_2O$ or $N_2$ at 10 °C, 20 °C, and 30 °C, respectively. From seeding (October) to regrowth (March), because the climate temperate is commonly low in HHHR, the wheat seedings grow slowly and N absorption is low accordingly; obviously, the N supply is far more than the N demand for wheat seedlings. It has been proven that six months of overwintering is an important period of N loss [10,11]. When the spring comes, and the climate turns warm, the wheat seedlings regrow fast, and the available nitrogen (Na) in soils is usually insufficient. From June to October is maize cultivation season, characterized by high temperatures and rain, a higher $NO_3^-$ level is probably the main cause of nitrogen loss and denitrification. Therefore, more topdressing fertilizers have to be applied in order to achieve a continuous desired yield, which certainly will increase both material cost and labor cost, as well the environmental risk. In the future, increased yield that relies on increased N inputs would likely result in larger reactive nitrogen (Nr) losses per unit of output; only increased yield driven by improved N use efficiency (NUE) would be expected to reduce Nr emissions per unit of output [7–10]. Therefore, to achieve yield and environmental security, closing the gap between actual and attainable NUE should be as important as closing yield gaps.

Fortunately, it has been found that urease inhibitors could effectively inhibit urea hydrolyzed rapidly in soils, so that urea might be held in an effective state as possible, and longer; nitrification inhibitors could inhibit $NH_4^+$ transforming into $NO_3^-$ and thus reduce the leaching loss of N. However, few studies have been conducted in field trials. Therefore, applying the combination of urease and nitrification inhibitors with basal fertilization would be hypothesized as a dual effective synergist for reducing N loss and pollution [12,13].

Here, this study will aim to explore the space-time changes and crop yield under different combinations of urease/nitrate inhibitors with basal fertilization in a wheat/maize rotation system, which intends to provide theoretical basis and technical support for scientific fertilization and environmental protection.

## 2. Materials and Methods

### 2.1. Experimental Field

Experiments were carried out at a seed-multiplication farm (E 116°89′–16°90′, N 35°29′–5°30′) located in Shiqiang Town, Zoucheng City, Shandong Province from October 2018 to October 2020. The region is a typical temperate monsoon climate, which features four distinct seasons. The yearly precipitation is 684.8 mm, concentrated from June to August, and accompanied by the maximum temperature 37.8 °C and the min temperature −13.7 °C. Field soils are brown soil, and the pH is 7.09; electrical conductivity (EC) is 130.3 Ms cm$^{-1}$. Nutrients contained in soils are as follows: organic matter, 16.26 g kg$^{-1}$; total nitrogen, 889.6 mg kg$^{-1}$; alkali-hydrolyzable N, 91.38 mg kg$^{-1}$; ammonia nitrogen ($NH_4^+$) 39.85 mg kg$^{-1}$; nitrate nitrogen ($NO_3^-$), 43.28 mg kg$^{-1}$; available P, 41.57 mg kg$^{-1}$; and available K 163.8 mg kg$^{-1}$.

*2.2. Materials and Experimental Design*

2.2.1. Materials

Wheat cultivar Tainong 18 (about 231 days growth period) and maize variety Zhengdan 958 (about 96 days growth period) were employed in this experiment. Wheat was sowed in mid-October and harvested in early June. Row spacing was 27 cm, and the sowing rate was 165 kg ha$^{-1}$. Maize was sown in mid-June and harvested in early October, row spacing was 33 cm, and the sowing density was 72,000 plants ha$^{-1}$.

Grade (N-P$_2$O$_5$-K$_2$O) of tested compound fertilizer was 19-18-18, urea-N and NH$_4^+$ account for 89.5% and 10.5% of total N, respectively. The N, P, and K sources of compound fertilizer was urea (N 46%), monoammonium phosphate (MAP) (N: 14%; P$_2$O$_5$: 46%) and potassium sulfate (K$_2$O: 51%), respectively. All these fertilizer sources were produced by the Agricultural Fertilizer Industry, China. The used urease inhibitors included hydroquinone (HQ) and N-(N-butyl) thiophosphoric triamide (NBPT), and nitrification inhibitors included dicyandiamide (DCD), 3,4-dimethylpyrazole phosphate (DMPP), and Nitrapyrin.

2.2.2. Experimental Design

Seven treatments were designated: (1) CK, conventional fertilizer application rate; (2) A1, HQ + DCD; (3) A2, HQ+ DMPP; (4) A3, HQ + Nitrapyrin; (5) A4, NBPT + DCD; (6) A5, NBPT + DMPP; and (7) A6, NBPT + Nitrapyrin. Treatments of A1–A6 were applied the same fertilizer as CK and in the same way, except for urease/nitrification inhibitors. Simultaneously, CK1 was conducted with same P and K fertilizers but with N deficiency, only aiming to calculate the apparent nitrogen recovery efficiency (ANRE) and nitrogen agronomic efficiency (NAE). Based on preliminary pot experiment and references [11,12], the appropriate amount of HQ and NBPT was 1.5% and 0.3% of total pure N; The appropriate amount of DCD, DMPP, and Nitrapyrin was 4%, 0.5%, and 0.25% of total pure N.

As local cultivation technical procedures recommended, in wheat season, the fertilizer amount was 225 kg N ha$^{-1}$, 120 kg P$_2$O$_5$ ha$^{-1}$, and 105 kg K$_2$O ha$^{-1}$; 100% K, 87.5% P, and 50% N were applied as basal fertilizer in the form of compound fertilizer, and the remaining 12.5% P and 50% N were top-dressed in the form of MAP and urea respectively in March of next year. In maize season, basal fertilizers and urease/nitrification inhibitors were applied simultaneously together with seeds through professional machine. The slow-released urea, NH$_4$H$_2$PO$_4$, and K$_2$SO$_4$ were used one-off as basal fertilizer, and the amount was 255 kg N ha$^{-1}$, 45 kg P$_2$O$_5$ ha$^{-1}$, and 60 kg K$_2$O ha$^{-1}$. In each treatment, each time basal fertilizers and corresponding urease/nitrification inhibitors were fully mixed, broadcasted, and plowed into field.

The field experimental plot was 50 m$^2$ (5 m width, 10 m length) and spaced 2 m apart. The plot was randomly complete block design, and each treatment included three field replications. Agricultural cultivation management was the same as local conventional practice.

*2.3. Sampling and Measurement*

2.3.1. Soil and Plant Samples' Collection

Soil and plant samples were collected at wheat seeding, greening, heading, and mature stages and at maize jointing, tasseling, and maturing stages. Soils were sampled following 'W-type' 10-point sampling method by a stainless-steel drill, and soils at 0–20 cm, 20–40 cm, and 40–60 cm depths were collected from the space between wheat rows and from space between maize individuals of a row. Soil samples from 10 points were mixed, and 300 g was transferred to the lab. Part of fresh soil sample was used to measure NH$_4^+$ and NO$_3^-$ contents; the left soil sample was air-dried, ground, sieved, and preserved for use.

At wheat mature stage, the one-meter-two-row method was employed to estimate yield components. Meanwhile, 10 plants were randomly sampled from yield estimation point and separated as grain, glume, and stem, and leaf parts to weight. The rest wheat was harvested for yield measurement. As for maize, 10 plants were randomly harvested from each plot, separated as ear and straw parts, air-dried, and weighted. Then grain yield

and straw yield per unit area were calculated. The sampled wheat and maize grains and straws were dried at 80 °C, ground, sieved, and subpackaged for N measurement.

### 2.3.2. N Evaluation of Soil and Plant Samples

Soil samples were firstly extracted using 2 mol $L^{-1}$ KCl solution (1 soil:5 water, *v/v*). $NH_4^+$ and $NO_3^-$ contents were analyzed by a continuous flow analysis (*TRAACS 2000,* Branand-luebbe, Norderstedt, Germany), and available N was determined by diffusion method.

Straws and grains were digested by $H_2SO_4$-$H_2O_2$ at first, and then the Kjeldahl method was used to determine N content [14].

### 2.3.3. N Efficient Index Calculation

Four N efficiency indexes were calculated as follows [15]:

Apparent nitrogen recovery efficiency (ANRE, %) = (Aboveground N amount of fertilizer treatment − Aboveground N amount of CK)/N application amount (kg) × 100%

Nitrogen agronomic efficiency (NAE, %) = (Yield of fertilizer treatment (kg) − Yield of CK (kg))/N application amount (kg) × 100%

Nitrogen harvest index (NHI, %) = Grain yield (kg)/Total biomass (kg) × 100%

Nitrogen partial factor productivity (NPFP) = Yield of N treatment (kg)/N application amount (kg).

### *2.4. Data Analysis*

The data in dynamic figure and column chart about nitrogen in soil were from 2019–2020. The crop yield and nutrient accumulation were the mean value of two rotations from 2018–2020. Office 2017 and Origin (OriginLab, Northampton, MA, USA) were performed for data processing and plotting. The least significant difference (LSD) was used for multiple comparisons between different treatment means.

## 3. Results and Analysis

### *3.1. Effect of Different Combinations of Urease and Nitrification Inhibitors on Soil $NH_4^+$ Level*

$NH_4^+$ content at 0–20 cm soil layer showed great fluctuation (Figure 1A), changing from 38.72–45.75 mg $kg^{-1}$ at 30 day after fertilization to 14.04–19.11 mg $kg^{-1}$ at the wheat mature stage. However, the change in $NH_4^+$ was gradually flatted in maize season (Figure 1B). Soil $NH_4^+$ fell to the lowest level till maize mature stage. The dynamic change indicated that main $NH_4^+$ loss occurred in the overwintering period of wheat (Figure 1A). The supplement of urease/nitrification inhibitors to basal fertilizers could increase soil $NH_4^+$, but $NH_4^+$ varied greatly among different treatments. The $NH_4^+$ content of A1 and A6 treatment was maintained at a higher level trend from the wheat seedling stage to the maize tasseling stage (Figure 1). At the maize harvest stage, all treatments supplied with urease/nitrification inhibitors had slightly lower $NH_4^+$ content than CK, but the difference was not statistically significant (Figure 1B).

All treatments supplied with urease/nitrification inhibitors showed higher $NH_4^+$ content than CK at 0–60 cm soil layer at the wheat harvest stage (Figure 2). When rainfall was rich in June and July, however, the content of $NH_4^+$ at the 40–60 cm soil layer was almost the same level as or even higher than at 0–20 cm soil layer under all treatments except for CK. Especially, the $NH_4^+$ content at the 40–60 cm soil layer under CK had the highest value among all treatments at wheat seeding stage, but it was decreased to the lowest value at wheat and maize mature stages. At maize mature stage, all treatments supplied with urease/nitrification inhibitors also presented higher $NH_4^+$ content than CK at all soil layers, indicating that urease/nitrification inhibitors were able to decrease nitrification effectively and continuously, but promote the occurrence of leaching.

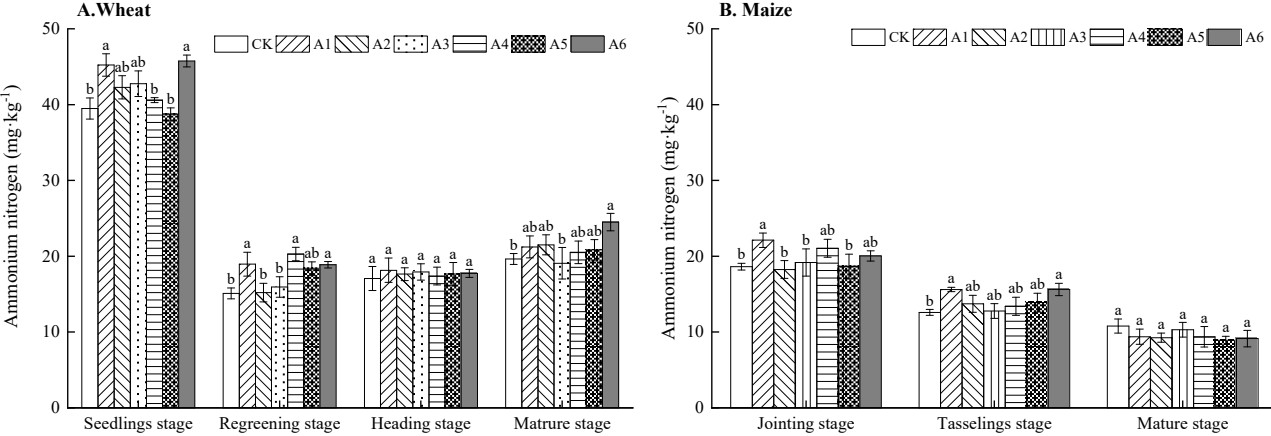

**Figure 1.** $NH_4^+$ content at 0–20 cm soil depth under different combinations of urease/nitrification inhibitors. The different lower letters indicate significant difference at $p < 0.05$.

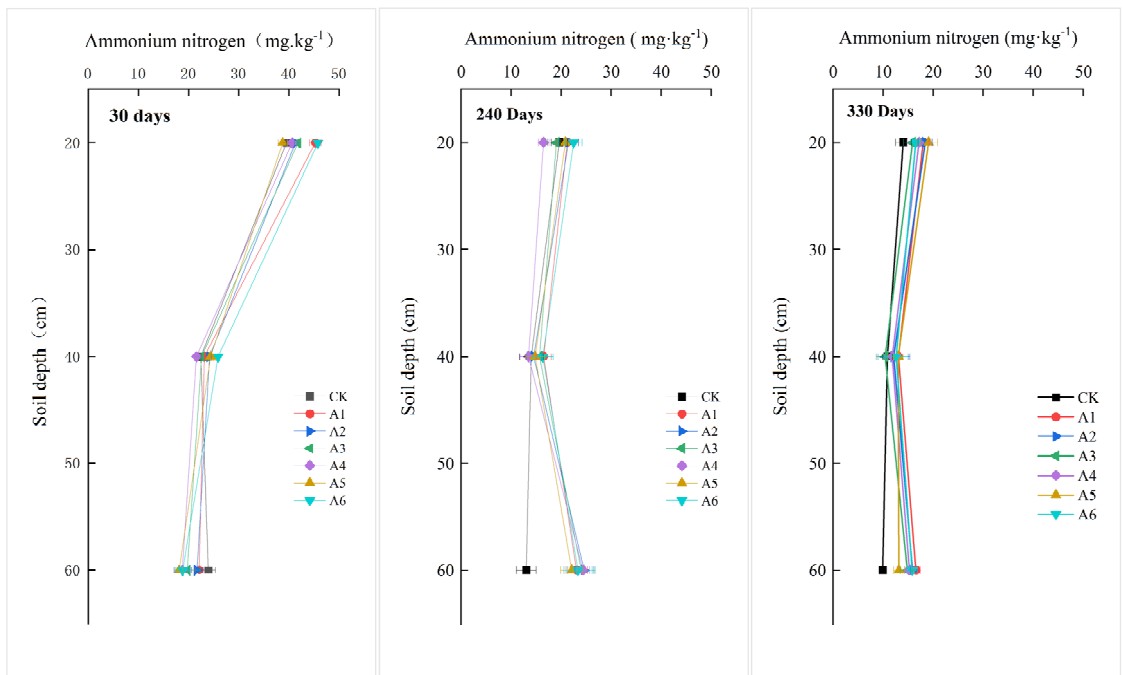

**Figure 2.** The temporal and spatial variation of ammonium in soil under different urease/nitrification inhibitor combinations. 30 days, 240 days, and 330 days indicated seedling stage, wheat harvest stage, and maize harvest stage; same below. The different lower letters indicate significant difference at $p < 0.05$.

### 3.2. Effects of Different Combinations of Urease/Nitrification Inhibitors on Soil $NO_3^-$

The $NO_3^-$ was obviously higher at the seedling stage than other stages at the 0–20 cm soil layer, and the loss rate was nearly 50% from seedling to green stages (Figure 3A). Soil $NO_3^-$ had the same change trend as $NH_4^+$. Greater loss of $NO_3^-$ also occurred in the overwintering period of wheat. The $NO_3^-$ content at 0–20 cm soil layer was significantly higher ($p < 0.05$) under A2 and A3 than CK, and the values were 52.68 mg kg$^{-1}$ and 52.26 mg kg$^{-1}$ under A2 and A3, respectively. The other treatments had values of $NO_3^-$ content with CK. Essentially, soil $NO_3^-$ content increased from the greening stage to the heading stage, but no significant difference was found between treatments supplied with urease/nitrification inhibitors and CK; soil $NO_3^-$ continued to reduce from the heading to mature stages and reached the minimum at the mature stage, when the difference between

combinations of urease/nitrification inhibitors and CK became larger (Figure 3B). However, only the difference between A1 and CK reached a significant level.

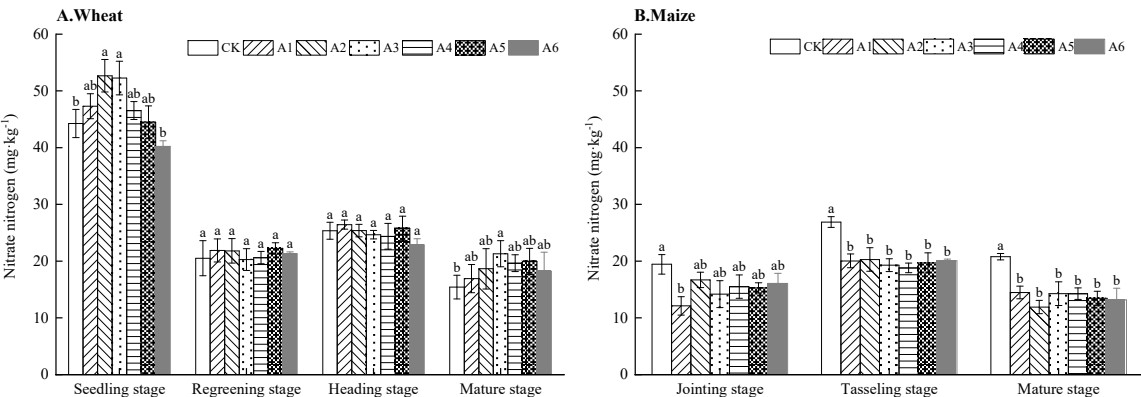

**Figure 3.** $NO_3^-$ level at 0–20 soil layer under different combinations of urease/nitrification inhibitors. The different lower letters indicate significant difference at $p < 0.05$.

Seen from Figure 3, at maize jointing and tasseling stages, when rainfall was more abundant, $NO_3^-$ moved downward from the surface to deeper soil layers (Figure 3B). Compared to CK, urease/nitrification inhibitor treatments significantly decreased $NO_3^-$ content at the 0–20 soil layer by 4.61–38.1%, and the difference was significant ($p < 0.05$).

In wheat/maize rotation period, $NO_3^-$ decreased gradually from the surface soil to a 60 cm soil depth, and as the soil layer descended, the $NO_3^-$ difference between treatments became smaller (Figure 4). At the wheat seeding stage (30 day after fertilization), $NO_3^-$ contents at the 0–20 cm soil layer of A2 and A3 were significantly higher than CK, but $NO_3^-$ content under A2 was significantly higher than CK only at the 20–40 cm soil layer; the other treatments were similar to CK in $NO_3^-$ content at the 0–20 cm and 20–40 cm soil layers. As for the 40–60 cm soil layer, the $NO_3^-$ content was decreased sharply, and it was approximately half of the value at upper soil layer. At the time points of wheat harvesting (240 days after fertilization) and maize harvesting (330 days after fertilizer application), $NO_3^-$ content was reduced greatly under all treatments, and most of the values were around 25 mg kg$^{-1}$ at the time point of wheat harvesting and 13 mg kg$^{-1}$ just before maize harvesting. With soil layer went down, $NO_3^-$ content was decreased, and the difference between treatments became smaller under all treatments except for CK. These indicated that nitrification inhibitors indeed worked at the 0–60 cm soil layer, which effectively prevented the conversion of $NH_4^+$ into $NO_3^-$ and reduced N leaching loss.

*3.3. Effects of Different Combinations of Urease/Nitrification Inhibitors on Soil Alkali-Hydrolyzable N*

In the entire wheat/maize growth period, the supplement of urease/nitrification inhibitors to basal fertilizer significantly improved the soil alkali-hydrolyzable N level at the 0–20 cm soil layer, and a greater effect occurred at the wheat seeding and greening stages (Figure 5A). The highest alkali-hydrolyzable N content at the 0–20 soil layer was observed under A4 at the wheat seeding stage and under A1 at the greening stage, but the two highest alkali-hydrolyzable N contents at the 0–20 cm soil layer were found under A1 and A6 at the mature stage. The alkali-hydrolyzable N content at the 0–20 cm soil layer showed the same changing trend as $NH_4^+$.

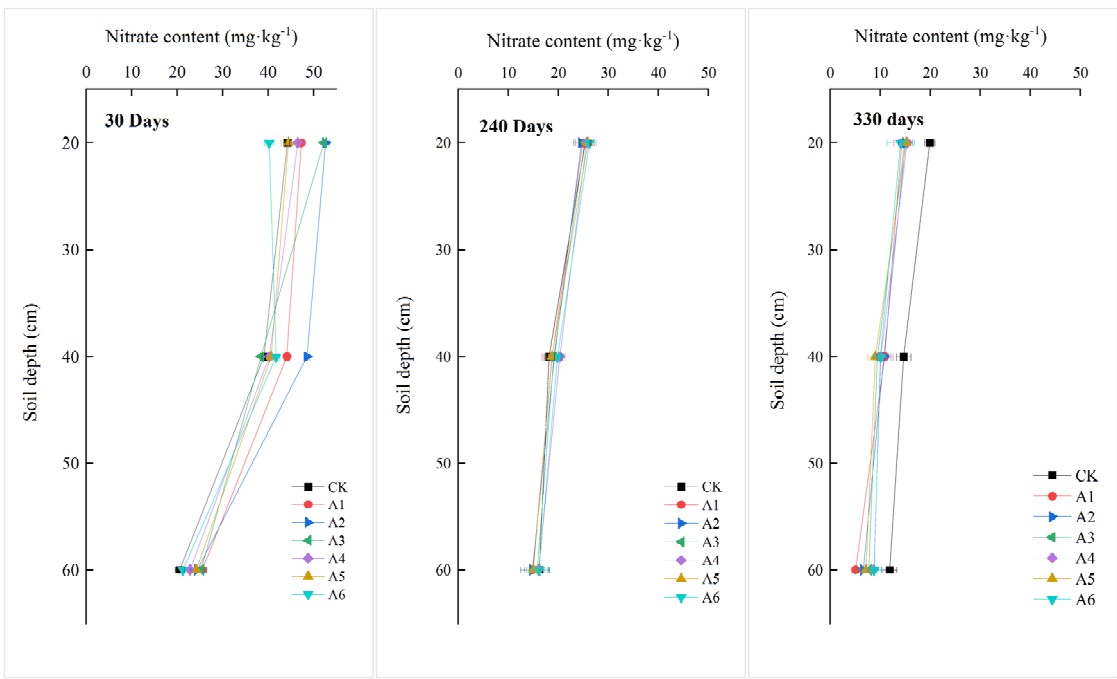

**Figure 4.** The temporal and spatial variation of $NO_3^-$ in soil under different combinations of urease/nitrification inhibitors.

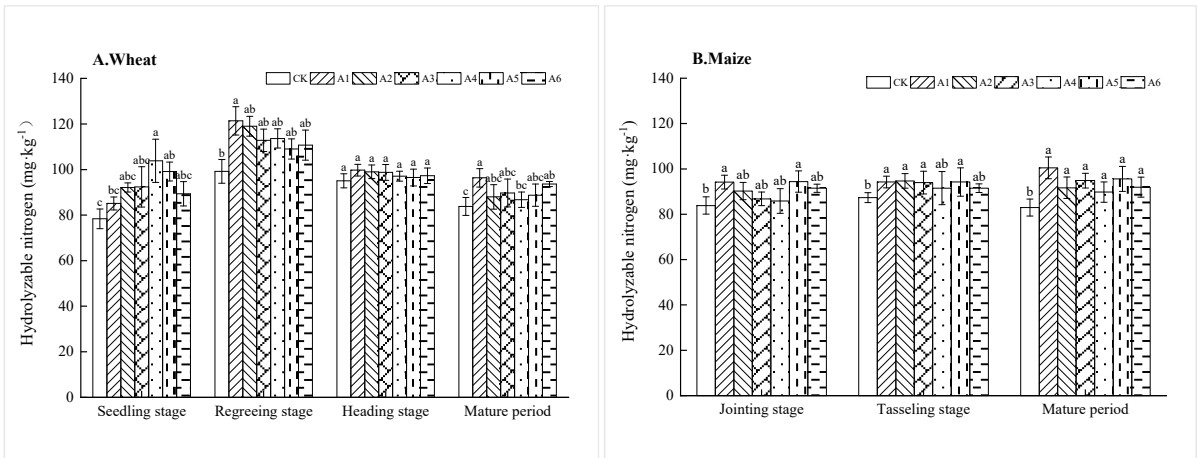

**Figure 5.** Alkali-hydrolyzable nitrogen content at the 0–20 soil layer under different combinations of urease/nitrification inhibitors. The different lower letters indicate significant difference at $p < 0.05$.

In the maize growth period, all treatments supplied with urease/nitrification inhibitors showed higher alkali-hydrolyzable N content, relative to CK, but the difference between treatments was smaller compared to that in the wheat season (Figure 5B). Higher alkali-hydrolyzable N contents were shown under A1 and A5 during the maize entire growth period, and it was 12.54 mg kg$^{-1}$ under A1 and 9.83 mg kg$^{-1}$ under A5 at the maize mature stage, which was 14.9% and 11.7% higher than CK ($p < 0.05$). The results indicated that the combination of HQ + DCD (A1) was capable of inhibiting the convention of urea and $NH_4^+$ into $NO_3^-$, thus steadily, continuously maintaining higher N status at the 0–20 cm soil layer.

The content of alkali-hydrolyzable N at 0–20 cm soil layer was within the range of 80–100 mg kg$^{-1}$, which was stable comparing to deeper soil layers, and the values were higher than those at deeper soil layers; the contents of alkali-hydrolyzable N at the 20–40 cm and 40–60 cm soil layers varied little at wheat seedling and mature stages but were obviously reduced after maize was harvested (Figure 6). During the entire experimental

period, alkali-hydrolyzable N content at all soil layers under all treatments supplied with urease/nitrification inhibitors was higher than CK, but the differences in available N between treatments became narrower at the deeper soil layer. Alkali-hydrolyzable N also leached, but the leaching degree was weak relative to $NH_4^+$ and $NO_3^-$.

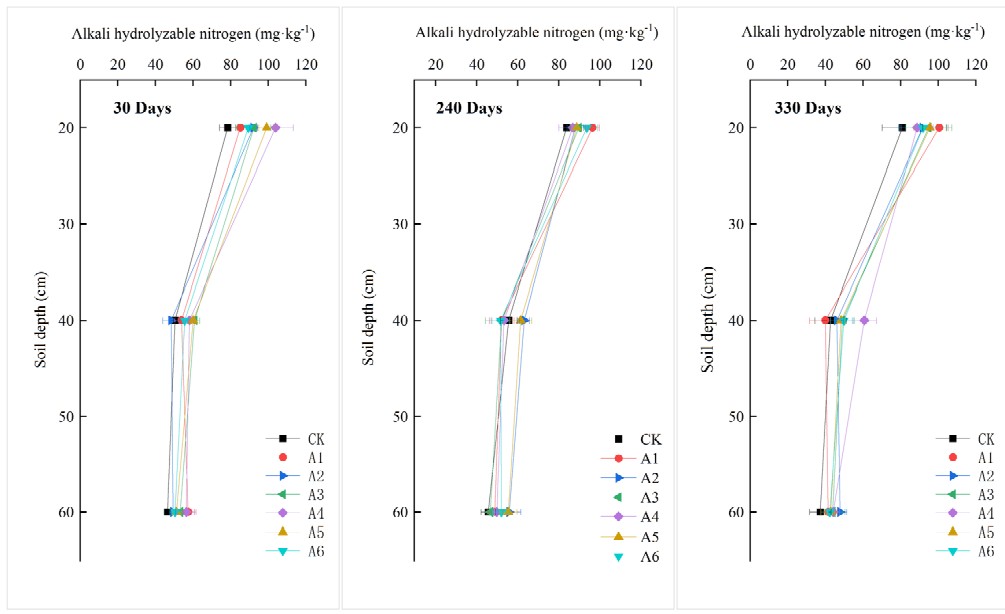

**Figure 6.** The temporal and spatial variation of alkali-hydrolyzable nitrogen in soil under different urease/nitrification inhibitor combinations.

### 3.4. Effects of Different Combinations of Urease/Nitrification Inhibitors on Soil Total N

In the wheat growth period, total N content at the 0–20 cm soil layer was slightly higher under all treatments than CK, but the difference did not reach a significant level in statistics except for A5 at the wheat seeding stage (Figure 7A). In the maize season, A5 showed the highest total N level (1.003 g kg$^{-1}$) at the jointing stage, while A1 achieved the highest total N level (1.012 g kg$^{-1}$) at the tasseling stage (Figure 7B). However, there was no significant difference between treatments ($p > 0.05$).

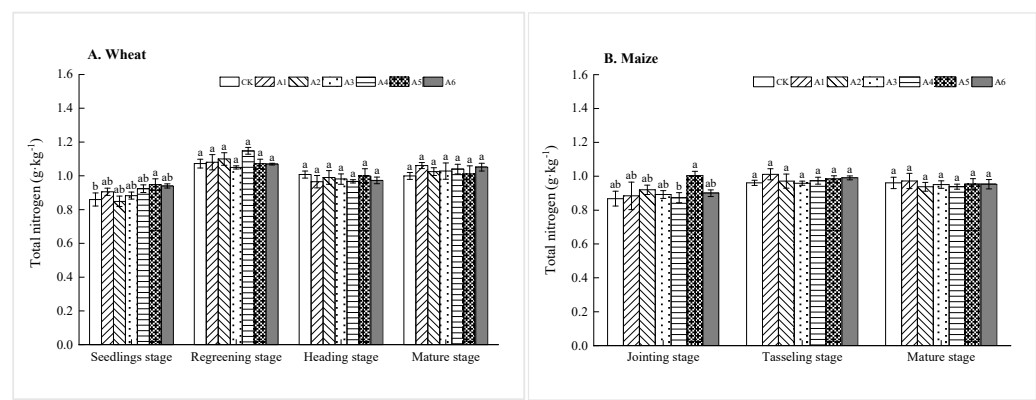

**Figure 7.** The total N content at 0–20 soil layer under different combinations of urease/nitrification inhibitors. The different lower letters indicate significant difference at $p < 0.05$.

### 3.5. Effects of Different Combinations of Urease/Nitrification Inhibitors on Yield and Biomass in Wheat and Maize

Urease/nitrification inhibitors significantly ($p < 0.05$) improved the biomass (including grain and straw) of both wheat and maize (Table 1). Among these treatments, A1 achieved the highest grain yield of 10047 kg ha$^{-1}$, followed by A3 at 9412 kg ha$^{-1}$, which were 16.3%

and 8.90% higher than CK. Accordingly, A1 and A3 were the two treatments achieving the most wheat straws, which were 14.6% and 18.1% more than CK. In the maize season, A1, A2, and A5 increased grain yield by 19.3%, 21.6%, and 19.8%, respectively, when compared to CK. The highest total grain yield and the highest total biomass were both found in A1, which were 19.9% and 15.4% higher than CK; the second highest biomass was achieved in A3, which was 13.5% more than CK. A1 and A3 obtained an increased yield in both wheat and maize seasons.

**Table 1.** Grain and straw production of wheat and maize under different urease/nitrification inhibitor combinations (kg ha$^{-1}$). Different letters within a column indicated significant difference at $p < 0.05$, the same as below.

| Treatment | Wheat | | Maize | | Total |
| --- | --- | --- | --- | --- | --- |
| | **Grain** | **Straw** | **Grain** | **Straw** | **Biomass** |
| CK | 8637 b | 12,285 b | 9421 b | 10,253 c | 40,596 e |
| A1 | 10,047 a | 14,083 a | 11,240 a | 11,493 b | 46,863 a |
| A2 | 8936 ab | 11,573 b | 11,471 a | 12,355 a | 44,335 b |
| A3 | 9412 a | 14,501 a | 10,916 a | 11,259 b | 46,088 a |
| A4 | 8927 ab | 12,702 ab | 10,181 ab | 10,747 c | 42,557 d |
| A5 | 8823 ab | 12,252 b | 11,290 a | 11,171 b | 43,536 c |
| A6 | 8933 ab | 12,159 b | 10,598 ab | 11,235 b | 42,925 d |

*3.6. Effects of Different Combinations of Urease/Nitrification Inhibitors on the N Intake by Plants of Wheat and Maize*

All treatments supplied with urease/nitrification inhibitors greatly increased N accumulation of grains and straws in wheat and maize (Table 2). In wheat season, the highest N accumulation of grain was found under A1, followed by A3, which was 27.7% and 15.9% higher than CK; the two highest values of N accumulation in straw were also found under A1 and A3, which were 29.9% and 32.5% higher than CK. In the maize season, when compared to CK, A4 increased grain N accumulation by 15.6% and increased straw N accumulation by 38.7%. The higher increase rate of straw N accumulation under A4 might be due to the higher straw N content. However, grain and straw N accumulation under A2 that had the highest grain yield in maize were 1.21% and 1.79%, respectively, which were lower than any other treatments. At harvest stage, wheat accumulated more N element in grains than in straws. By contrast, maize accumulated more N in straws than in grains, but both wheat and maize had a higher increased degree in grain N accumulation than in straw N accumulation.

**Table 2.** N accumulation in wheat and maize under different combinations of urease/nitrification inhibitors (kg ha$^{-1}$).

| Treatment | Wheat | | Maize | | The |
| --- | --- | --- | --- | --- | --- |
| | **Grain** | **Straw** | **Grain** | **Straw** | **Total** |
| CK1 | 176.2 d | 110.67 d | 112.8 b | 165.1 e | 564.7 c |
| A1 | 225.1 a | 143.7 a | 146.2 a | 202.3 c | 717.2 a |
| A2 | 193.0 bc | 114.6 d | 139.0 a | 221.2 a | 667.7 b |
| A3 | 204.2 b | 146.5 a | 135.1 a | 186.9 d | 672.7 b |
| A4 | 191.9 bc | 128.3 b | 130.4 ab | 228.9 ab | 679.6 b |
| A5 | 187.1 c | 128.7 b | 142.5 a | 221.2 ab | 679.4 b |
| A6 | 201.9 b | 121.6 c | 133.4 ab | 215.7 b | 672.6 b |

*3.7. Effects of Different Combinations of Urease/Nitrification Inhibitors on N Use Efficiency in Wheat and Maize*

All treatments supplied with urease/nitrification inhibitors significantly improved the ANRE in wheat and maize (Table 3, $p < 0.05$). Specifically, ANRE was increased by 28.2–111.0% in the wheat season and increased by 51.4–100.0% in maize season. The ANRE values in the wheat season were basically higher than in the maize season. In wheat growth season, A1 had the highest ANRE of 64.9%, which was 111.0% higher than CK; A3 had the second highest ANRE of 57.4%, the value being 86.6% higher than CK. For maize production, the highest ANRE (67.5%) was found under A5, which was 100.0% higher than CK, followed by A1, and its ANRE was improved by 95.9%, when compared to CK.

**Table 3.** N use efficiency in wheat and maize under different combinations of urease/ nitrification inhibitors.

| Treatment | ANRE (%) | | NAE (kg kg$^{-1}$) | | NHI (%) | | NPFP (%) | |
|---|---|---|---|---|---|---|---|---|
| | **Wheat** | **Maize** | **Wheat** | **Maize** | **Wheat** | **Maize** | **Wheat** | **Maize** |
| CK | 30.75 c | 33.76 c | 6.238 d | 5.472 d | 61.44 a | 40.60 a | 35.99 b | 37.09 c |
| A1 | 64.89 a | 61.55 a | 12.11 a | 12.63 a | 61.04 a | 41.96 a | 41.86 a | 44.25 a |
| A2 | 39.43 b | 66.13 a | 7.48 3c | 13.54 a | 62.75 a | 38.59 ab | 37.23 b | 45.16 a |
| A3 | 57.39 a | 51.11 b | 9.467 b | 11.36 ab | 58.24 a | 41.95 a | 39.22 ab | 42.98 ab |
| A4 | 44.69 b | 65.82 a | 7.446 c | 8.46 c | 59.94 a | 36.30 b | 37.20 b | 40.08 b |
| A5 | 42.80 b | 67.52 a | 7.013 c | 12.83 a | 59.25 a | 39.18 ab | 36.76 b | 44.45 a |
| A6 | 46.04 b | 61.80 a | 7.471 c | 10.11 b | 62.41 a | 38.21 ab | 37.22 b | 41.72 ab |

All treatments supplied with urease/nitrification inhibitors were improved in NAE in both wheat and maize seasons. The increased percentage of NAE was 12.4–94.1% in the wheat season; the two highest NAE, 12.11 kg kg$^{-1}$ found under A1 and 9.48 kg kg$^{-1}$ under A3, were 94.1% and 51.8% higher than CK. NAE was increased by 54.6–147.4% in maize season. The highest NAE (147.4%) was observed under A2, 147.4% higher than CK; the second highest NAE (12.8%) was found under A5, which was 134.5% higher than CK.

All treatments with urease/nitrification inhibitors had a similar nitrogen harvest index (NHI) in wheat season but not in maize season. In maize season, the NHI under A4 was 10.6% lower than CK, and the difference reached a significant level ($p < 0.05$), which might be related to the later maturity of this treatment. Generally, the NHI of wheat was 20% higher than maize, and the transport of substances in wheat was stronger than in maize, but the biomass of wheat was always lower than that of maize.

In the wheat season, all treatments with urease/nitrification inhibitors (A2–A7) showed improved NPFP, but the difference was significant ($p < 0.05$) only among A1, A3, and CK. In the maize season, A2–A7 had higher NPFP than CK, and the increased percentage ranged from 8.06% to 21.8%. A2 showed the highest NPFP (45.2%), followed by A5 (44.5%).

## 4. Discussion

### 4.1. Relationship between Urease/Nitrification Inhibitors and Soil N Levels

$NH_4^+$ and $NO_3^-$ were the most available inorganic nitrogen resource for crops, because they could be quickly absorbed and assimilated by plants when crops were in the N deficiency state. Alkali-hydrolyzable N was also a type of available N, which included inorganic N (mainly $NH_4^+$ and $NO_3^-$) and organic N (amino acids, amides, and hydrolyzable proteins), which could reflect soil N availability [16]. The presented study showed that some alkali-hydrolyzable N could move down from the surface to deeper soils when heavy rainfall occurred, but the supplement of urease/nitrification inhibitors to the basal fertilizer could relieve the effect.

Relative to alkali-hydrolyzable N, $NH_4^+$ and $NO_3^-$ seemed greatly active and movable, and the values of $NH_4^+$ and $NO_3^-$ at the surface soils at the wheat seedling stage were nearly twice of those at other stages. This might because, on the one hand, overwintering water (mainly snow) resulted in the leaching of surface $NH_4^+$ and $NO_3^-$; on the other hand, chemical fertilizer accounted for most in the basal fertilizers at the wheat seedling stage, and the high concentrations of $NH_4^+$ and $NO_3^-$ in the surface soil intensified N leaching loss [17]. The $NH_4^+$ at the 20–40 cm soil layer fluctuated in a similar trend, but $NH_4^+$ differences between treatments were narrower than that at the 0–20 cm soil layer. The greater changes in $NH_4^+$ content at the upper soil layer were likely due to leaching occurring along with rainfall and irrigation, and (or) due to $NH_4^+$ volatilization caused by denitrification. Thus, the difference in $NH_4^+$ at the 0–20 soil layer between different treatments, became narrower at both wheat and maize mature stages, and in the meantime, the differences between upper and lower soil layers also became narrower.

Although all combinations of urease/nitrification inhibitors increased soil total N content, the effect was much weaker (2.28–6.41%). The field experiments that continued from 2018 to 2020 confirmed that some $NH_4^+$ in the surface soil leached. Through comparing, $NH_4^+$ and $NO_3^-$ at the 0–20 cm soil layer leached most in June and July, but $NH_4^+$ and $NO_3^-$ at the 20–40 cm soil layer was increased accordingly. This result was inconsistent with the conventional viewpoint that it was not prone to leach for $NH_4^+$ This result might result from the intensive rainstorm in July 2019 at the experimental location. The supplement of urease/nitrification inhibitors to basal fertilizer positively affected the formation of $NH_4^+$ in surface soil, which greatly enhanced the risk of available N leaching at the maize growth period, when it was hot and accompanied with abundant rainfall. However, urease/nitrification inhibitors restrained the generation of $NO_3^-$ and to some degree reduced the leaching loss of $NH_4^+$ and $NO_3^-$. The results were consistent with previous studies which found that the supplement of DMPP increases $NH_4^+$ content while it decreases $NO_3^-$ content in leachate [18–21].

In both wheat and maize seasons, urease/nitrification inhibitors increased soil $NH_4^+$ content but inhibited the $NO_3^-$ content at 0–20 cm soil layers in maize growth period. This study illustrated that the decrease in $NO_3^-$ at surface soil reduces denitrification [22]. Although the addition of urease/nitrification inhibitors in basal fertilizer reduced to some extent, the $NO_3^-$ content at 0–60 cm soil layer, the increases in maize yield and fertilizer use efficiency indirectly proved that the urease/nitrification inhibitors' combinations effectively maintained available N in root zone.

*4.2. Relationships among Urease/Nitrification Inhibitors, Yield and N Accumulation in Wheat and Maize*

The urease/nitrification inhibitor also played a positive role in N intake, N accumulation, and N assimilation of plants. Compared with conventional fertilizers, the supplement of urease/nitrification inhibitors to basal fertilizer increased the tillers of wheat, and kernel number per ear and 1000-kernel weight of maize (data not shown), improved final grain yield and straw yield in wheat and maize, and enhanced the total biomass. However, the increase in both grain and straw yields was not completely synchronous. Wheat seedlings required little total N, but soil N concentration was the highest in early growth stage. The higher level of $NH_4^+$ and $NO_3^-$ at wheat seedling stage indirectly proved that the excessive supplement of basal fertilizer had no significant effect on the growth and development of plants, while topdressing in time and in appropriate amount played a key role in N accumulation, yield, and N use efficiency [22]. The result was also in line with the current idea of deferring N application and balancing fertilizers [16,23].

According to N accumulation in wheat and maize, the different combinations of urease/nitrification inhibitors promoted more N accumulation in straws than in kernels, and this situation was more obvious in maize. This may be derived from the earlier harvest of A4 treatment in 2019 [24,25], and when putting off harvest time until 10 days later in 2020, maize grain yield was greatly uncovered, and straw yield was reduced.

*4.3. Relationship between Urease/Nitrification Inhibitors and N Use Efficiency*

For field crops, the ANRE was closely related to N fertilizer form, N fertilizer application period, and N fertilizer release, etc. [26]. In the Huang-Huai-Hai region, due to the low mechanization level and high cost of topdressing fertilizer, it was still customary to apply 60–90% of N fertilizer into the soil along with basal fertilizer in wheat/maize rotation. Urea is the main N source in compound fertilizer, and it takes only 12 days for urea to be completely hydrolyzed even at the soil temperature of 10 °C [27]. The temperature at the wheat seeding stage averaged 5–14 °C and lasted nearly for 120 days, so hydrolysis or ammonization of urea and nitrification of $NH_4^+$ were inevitable, and then leaching or denitrification of N occurs. In this study, the supplement of urease/nitrification inhibitors to basal fertilizer significantly improved the ANRE of wheat and maize. The optimal combination is HQ + DCD, in which ANRE was up to 64.9% in wheat season and 67.5% in maize season, and the ANRE values were 111% and 100% more than CK. The increase in ANRE was due to the synergistic increases of grain yield, straw yield, and N absorption, which effectively enhanced the biological immobilization of soil N, and thus directly or indirectly reducing N loss.

NAE indicated the grain output per unit N; NHI was the proportion of grain yield to total biomass; NPFP was the grain yield of N unit applied. These three parameters of N were affected by many factors, including soil N level, soil texture, pH, C/N, and harvest time [26,28]. In this study, all the combinations of urease and nitrification inhibitors greatly improved NAE, NHI, and NPFP. All treatments supplied with urease and nitrification inhibitors had similar effect on the three parameters of wheat, and the improving effects were much more obvious than that on soil N and yield. The improving effects of urease/nitrification inhibitors on three N parameters varied significantly in the maize season. In particular, the combination of HQ + DCD (A2) had the lowest NAE in wheat season while achieves the highest value in maize season. Probably this was mainly due to the early harvest of maize, when straws contained more N element. The NAE, NHI, and NPFP in the maize season were all higher than in the wheat season, but the distribution rate of N to kernels was smaller in maize season than in wheat season. So, the HI of maize was 20% less than of wheat. According to the relationships among grain yield, biomass, and soil nutrients, crop biomass increased with the increase in soil available N, but crop yield was not consistent with the change trend of biomass. This suggested that a balanced supply of soil nutrients contributed more to yield than a single nutrient [29,30].

N nutrient was crucial for crops' growth and development and played a key role in yield formation of crops. More than 95% of soil N was organic N, and it must be transformed into $NH_4^+$ or $NO_3^-$ by microbial mineralization for plant use. In the rapid growth period or the maximum nutrition efficiency period of crops, the available N from soil was much less than real requirement of crops, so fertilization was essential for yield [1,22].

Based on the spatiotemporal changes of available N, $NH_4^+$, and $NO_3^-$ in soils, urease/nitrification inhibitors did not always increase nutrients in soil layer of roots, but extremely, significantly improved crop yield, and the increasing degree of yield and N efficiency was higher than that of soil available N. This phenomenon indicated that the N intake of plants were from both fertilizer and soils. The results of this study declared that the highest ANRE achieved under A1 was 111% more than CK. The increasing rate was generally higher than previous studies, which may be because (1) the traditional calculation method of ANRE ignored the stimulation or activation effect of fertilizer or exogenous synergist on soil nutrients, and ANRE could not represent the real NUE [31,32]; and (2) the contribution of urease/nitrification inhibitors to yield and NUE was the result of the effective regulation of urease/nitrification inhibitors on ammonization and nitrification of soil organic N. In this study, path analysis showed that alkali-hydrolyzable N had the highest correlation coefficient with grain yield ($R$ = 0.779), which positively affected the grain yield (path coefficient 4.455), and also affected organic matter ($R$ = 0.521) and soil total N content ($R$ = 0.466) (data not shown). Thus, in the wheat/maize rotation system combined with straw incorporation, the N in small organic molecules was the main form contributing to

grain yield. In sum, the combination of urease/nitrification inhibitors with basal fertilizers may show decreased soil $NH_4^+$ and $NO_3^-$ levels at some time, but it substantially reduced N loss [18], improved N supply capacity of soil [33–35], and finally achieved higher wheat and maize yield. However, some issues needed to be further explored, such as the contribution rate of urease/nitrification inhibitors to the effectiveness of organic soil N and urea N, and the contribution rate of fertilizer N and soil N to yield, and even the feasibility in slow-controlled release fertilizer. Predictably, the optimized and integrated combination of urease/nitrification inhibitors with basal fertilizers could not only reduce the nitrogen loss, promote efficient utilization of soil and fertilizer resources, but also greatly reduce environmental risks from crop management.

## 5. Conclusions

(1) The combination of urease/nitrification inhibitors with basal fertilizers could improve soil fertility, prevent alkali-hydrolyzable N, $NH_4^+$, and $NO_3^-$ from leaching, and increase available N level at a 0–20 cm soil layer, thus maintaining a higher level of alkali-hydrolyzable N and $NH_4^+$ during the entire growth period and promoting the absorption and transport of N nutrient from root to plant shoot. Urease/nitrification inhibitors also enhanced the final yield of wheat and maize, and the combination of 1.5% HQ, 4% DCD, and 0.25% Nitrapyrin was the optimal solution to improve grain yield, in which wheat yield was increased by 16.3% and maize yield was enhanced by 20.0%, when compared to CK. The second optimal solution was HQ + DCD, which showed the highest grain yield and achieves higher ANRE, NAE, and NPFP.

(2) The yield increase in wheat and maize resulting from application of urease/nitrification inhibitors was due to the improvement in total available N soil, but not so much due to the increase in $NH_4^+$, $NO_3^-$, or alkali-hydrolyzable N.

**Author Contributions:** Conceptualization, X.C. and Y.D.; methodology, J.W. (Jingquan Wang); software, J.W. (Jiahui Wang) and J.W. (Jingquan Wang); validation, Y.L. (Yun Li) and Y.L. (Yanhong Lou); formal analysis, X.C. and Y.D.; data curation, J.W. (Jiahui Wang) and J.W. (Jingquan Wang); writing—original draft preparation, J.W. (Jingquan Wang); writing—review and editing, X.C. and Y.D.; visualization and project administration, Y.Z.; supervision, X.C. and Y.D. All authors have read and agreed to the published version of the manuscript.

**Funding:** This research was funded by the Key Research and Development Program of Shandong Province (2021CXGC010801) and Natural Science Foundation of Shandong Province (ZR2021MC145).

**Institutional Review Board Statement:** Not applicable.

**Informed Consent Statement:** Not applicable.

**Data Availability Statement:** Not applicable.

**Conflicts of Interest:** The authors declare no conflict of interest.

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
