# Peer review of "Soil Available Nitrogen and Yield Effect under Different Combinations of Urease/Nitrate Inhibitor in Wheat/Maize Rotation System"

_agronomy, doi:10.3390/agronomy12081888_

Round 1

Reviewer 1 Report

The authors improved the MS, that is suitable to be accepted, however the experimantal design used for the study might be added to the work? RCBD or other ones?

Author Response

Modified, added relative content.

Reviewer 2 Report

thank you for giving me the opportunity to review the manuscript n. agronomy-1833227 entitled “EFFECTS OF DIFFERENT COMBINATIONS OF BASAL FERTILIZATION AND UREASE/NITRATE INHIBITOR ON CROP YIELD AND SOIL NITROGEN CONTENT IN A WHEAT/MAIZE ROTATION SYSTEM”. The manuscript aims to investigate the effects of urease/nitrification inhibitors applied in combination with basal fertilizers on soil N level and grain yield of wheat and maize crops. The topic is of interest and actual considering the need at worldwide level to reduce the application of mineral N fertilizer and the improvement of their use efficiency.

TITLE:

The title resulted a bit long, and, in my opinion, it should be reduced. Actually in the title it is preferable to avoid statements that start as “the effects of…”, “Influence of…) and other. I suggest revising the title making it more attractive. In addition based on the whole manuscript, the nitrogen is studied at different levels reporting only the soil nitrogen in the title is very limitative compared the work done.

ABSTRACT:

The first sentence of the abstract should highlight better the problem related to the reduced N use efficiency and the risk of nitrogen losses in the environment, with special focus on the loss and the problem that N losses could generate. In addition, by reading the first sentence, but also the title, the authors assume that wheat/maize rotation is widely adopted therefore more focus should be added to this practice.

Line 13-14: there is reported six N stabilizing combinations but there are reported two urease inhibitors and three nitrification inhibitors. In my opinion the description of the abstract should be improved to make more easy the experimental setup, therefore I suggest to review the treatment description.

In addition, I suggest to add the measurements performed in the study. The sentence reported in lines 14-16 “The treatments supplied with urease/nitrification inhibitors reduced, to some degree, the conversion rate of NH+ 4 into NO3 , and kept NH+ 4 content higher in surface soils for a longer time” is difficult to follow because it is missing a part of the story!

Line 16: what is CK? Please specify

Line 17: I suggest to report the main findings observed in the study, stating with “for example,…” is not significative. The abstract should be a part that attract the readers to read the whole manuscript.

Line 18: what is A1? Please specify

Line 19: in the sentence you report “entire growth period”, which? When? These information are missing in the abstract and make confusion for the readers.

In line 18 you are reporting the soil layer of 0-20 cm while in the line 28 the soil liyer was 0-40 cm, is it ok?

KEY WORDS:

The keyword “maize-wheat rotation” is already reported in the title, it is preferable to change with another one.

INTRODUCTION:

Line 37: please change gain with grain

Line 37: what do you mean with the term “fertilizer exits”, please could you change with another term more significative. In addition, this is not only related to low education of farmers, but generally the application of mineral N fertilizer is related to stress the nutrienlt plant uptake in order to produce more.

The introduction needs to be improved. It would be great to add a paragraph regarding the K fertilization effect on crops in general and specifically in rice by adding information reported in previous studies published. At this step the introduction section focus mainly on nitrogen application and based on the title it is not the focus of the study. Therefore, changes in title and introduction should be done to address correctly the information flux of the manuscript. In addition, in the introduction it is missing the hypothesis of the study and the objective. Please add these aspects, without it is very difficult for a reader to have a complete idea of the research.

Line 55-71: after the reading of this period I am convinced that the main problem is related to the N fertilization of wheat crop. So the maize crop is notfertilized? I think that as graminaceous species also the mais receive a lot of N fertilizers, therefore also the N fertilizer management contribute to the N losses and pollution. I suggest adding a period related to the maize cultivation and the main gaps on N fertilization in HHHR region.

I suggest to add clearly the objectives of this study, at this time, the last sentence make some confusion and it is not clear what are you study?

Line 79: I suppose there is a mistake “Na” should be changed with “N”.

MATERIALS and METHODS:

Lines 85-87: I suggest removing the first sentence, it is not significant here. It should be reported during the crop management.

Line 89: therefore, was the study performed at field condition and oin two consecutive growing season? I suggest to report these information more clear. If possible, for each growing season, please add a table where is reported the data of all agronomical practices adopted, with special focus on the N fertilizer applications (when and the amount) this help the reader to have a clear idea of the field conditions.

Line 91: if the term concentrated is reported, I am waiting also the period when the precipitation is concentrated i.e. from October to April.

Line 102: please add the information of row spacing and seed rate of maize crop, as reported for the wheat.

Line 107-109: the inhibitors are commercial products or are new products tested in this experiment?

Line 111: here is reported 7 treatments, in the abstract 6, please be consistent thoughout the manuscript.

I suggest to improve the subsection “Experimental design” with the wheat and maize management from an agronomical aspects and integrate with the table that I before suggested. This will help the readers to have a clear idea on the agronomical practices adopted in the study.

Line 135: so, no plant sample collections have been done in maize?

Lines138-139: here is reported soil collections in maize, but no plant samples have been collected? Is there a mistake? Please specify. In addition why soil samples in wheat have been collected between rows and in maize within rows? These aspects should be clarified.

Line 166: this section should be better described. I am not sure regarding the experimental design do you adopted for the statistical analysis. How the growin season was treated? These aspects should be clear for the readers.

RESULTS:

Figure 1: it is too small and it is difficult to read the letters, I suggest to report the figures vertically and increase the size. I am not sure (but small size) the unit of measure is correct, please check. What means the vertical bars? This should be clarified in the figure caption.

Line 178: 30 days after fertilization means that it corresponds with “seedling stage”? if so please specify.

Lines177-185: the two figures should be better discussed. Now there is confusion. I suggest to rewrite starting with the description of the figure 1a and then figure 1b. In this way it is more clear the results related to the soil NH4 content during the main crop stages.

Figure 2. It is very difficult to see the differences among the treatments, the figures should be improved. I suppose that these measurements are related to the maize crop, but it is not clear for me. In addition, in the figure is reported horizontal rows that should be explained in the figure caption.

Line 202: please avoid the term “For example”, here you are reporting the findings of your study, this need to be clear and concise.

Figure 3. The figure format should be improved, it is very small to be read.

Figure 4. The same consideration of Fig 2. 30, 240 and 330 days from what? This is not clear and could make confusion for a potential reader.

Table 1. I suggest removing doubled information. One way could be to report the harvest index of wheat and maize instead of biomass of each crop (it is the sum of the previous data), while in the total yield, I suggest removing grain and straw and leave the total biomass. The same could be done in Table 2.

In all tables remove the note and add the information in the caption.

DISCUSSION:

The application of organic N fertilizers contributes to reduce the N mineralization and thus N losses. Although it is not considered as experimental treatment in this study, I suggest adding some sentences related to the importance of organic fertilizers could play in the N cycling.

I suggest also to report more implication that the adoption of N inhibitors could have in the crop management and the benefits obtained from this experiment. Moreover, how these findings could be applied in the next study?

Author Response

Reply to the reviewer 2

TITLE:

The title resulted a bit long, and, in my opinion, it should be reduced. Actually in the title it is preferable to avoid statements that start as “the effects of…”, “Influence of…) and other. I suggest revising the title making it more attractive. In addition based on the whole manuscript, the nitrogen is studied at different levels reporting only the soil nitrogen in the title is very limitative compared the work done.

Reply: The title has been changed to "Soil available nitrogen and yield effect under different combinations of urease/nitrate inhibitor in wheat/maize rotation system”

ABSTRACT:

The first sentence of the abstract should highlight better the problem related to the reduced N use efficiency and the risk of nitrogen losses in the environment, with special focus on the loss and the problem that N losses could generate. In addition, by reading the first sentence, but also the title, the authors assume that wheat/maize rotation is widely adopted therefore more focus should be added to this practice.

Reply: The title and the first sentence have been modified, see line 2 and line 11

Line 13-14: there is reported six N stabilizing combinations but there are reported two urease inhibitors and three nitrification inhibitors. In my opinion the description of the abstract should be improved to make more easy the experimental setup, therefore I suggest to review the treatment description.

Reply: Modified, see line 13

In addition, I suggest to add the measurements performed in the study. The sentence reported in lines 14-16 “The treatments supplied with urease/nitrification inhibitors reduced, to some degree, the conversion rate of NH+ 4 into NO3 , and kept NH+ 4 content higher in surface soils for a longer time” is difficult to follow because it is missing a part of the story!

Reply: Because the reducing value is different among different treatment, it is a range or general trend. We think this sentence had better not modify.

Line 16: what is CK? Please specify

Reply: CK was specified in L2.2.2, line 118. Because the word number is limit, there is no detailed description in abstract.

Line 17: I suggest to report the main findings observed in the study, stating with “for example,…” is not significative. The abstract should be a part that attract the readers to read the whole manuscript.

Reply: Modified, deleted “for example”

Line 18: what is A1? Please specify

Reply: A1 was specified in L2.2.2, line 119-125. Because the word number is limit, there is no detailed description in abstract.

Line 19: in the sentence you report “entire growth period”, which? When? These information are missing in the abstract and make confusion for the readers.

Reply: The entire growth period means from sow seed to harvest, the time is a little different every year according to specific weather conditions.

In line 18 you are reporting the soil layer of 0-20 cm while in the line 28 the soil liyer was 0-40 cm, is it ok?

Reply: It is OK. the rising range in 0-20 cm is higher than 20-40cm, but both are improved.

KEY WORDS:

The keyword “maize-wheat rotation” is already reported in the title, it is preferable to change with another one.

Reply: Deleted “maize-wheat rotation”, added “increase production”, see line 32

INTRODUCTION:

Line 37: please change gain with grain

Reply: Modified, thanks!

Line 37: what do you mean with the term “fertilizer exits”, please could you change with another term more significative. In addition, this is not only related to low education of farmers, but generally the application of mineral N fertilizer is related to stress the nutrienlt plant uptake in order to produce more.

Reply: Modified to “the phenomenon of excessively applying fertilizer commonly exits due to low educational level of farmers or employees.”

The introduction needs to be improved. It would be great to add a paragraph regarding the K fertilization effect on crops in general and specifically in rice by adding information reported in previous studies published. At this step the introduction section focus mainly on nitrogen application and based on the title it is not the focus of the study. Therefore, changes in title and introduction should be done to address correctly the information flux of the manuscript. In addition, in the introduction it is missing the hypothesis of the study and the objective. Please add these aspects, without it is very difficult for a reader to have a complete idea of the research.

Reply: We think “add a paragraph regarding the K fertilization effect on crops in general” is not suitable, because our aim of this paper is about nitrogen.

Line 55-71: after the reading of this period I am convinced that the main problem is related to the N fertilization of wheat crop. So the maize crop is notfertilized? I think that as graminaceous species also the mais receive a lot of N fertilizers, therefore also the N fertilizer management contribute to the N losses and pollution. I suggest adding a period related to the maize cultivation and the main gaps on N fertilization in HHHR region.

Reply: Modified, see line 74-76.

I suggest to add clearly the objectives of this study, at this time, the last sentence make some confusion and it is not clear what are you study?

Reply: Modified, see line 90-92

Line 79: I suppose there is a mistake “Na” should be changed with “N”.

Reply: Modified, see line 87

MATERIALS and METHODS:

Lines 85-87: I suggest removing the first sentence, it is not significant here. It should be reported during the crop management.

Reply: Modified, see line 96

Line 89: therefore, was the study performed at field condition and oin two consecutive growing season? I suggest to report these information more clear. If possible, for each growing season, please add a table where is reported the data of all agronomical practices adopted, with special focus on the N fertilizer applications (when and the amount) this help the reader to have a clear idea of the field conditions.

Reply: Modified. Yes, there res two growing season from 2018-2020. The cultivation and fertilization information are line 125-129

Line 91: if the term concentrated is reported, I am waiting also the period when the precipitation is concentrated i.e. from October to April.

Reply: Modified, see line 99-100

Line 102: please add the information of row spacing and seed rate of maize crop, as reported for the wheat.

Reply: Added “the row spacing and seed rate of maize crop”, see line 128-129

Line 107-109: the inhibitors are commercial products or are new products tested in this experiment?

Reply: The inhibitors are commercial products

Line 111: here is reported 7 treatments, in the abstract 6, please be consistent thoughout the manuscript.

Reply: In the abstrac, it is “Six N stabilizing combinations”, here is “7 treatments” conluding CK.

I suggest to improve the subsection “Experimental design” with the wheat and maize management from an agronomical aspects and integrate with the table that I before suggested. This will help the readers to have a clear idea on the agronomical practices adopted in the study.

Reply: Because the number of table in an manuscript is limited to three, it is not accorded to list excessive tables. And more the agronomical practices are not proper for table, the authors think.

Line 135: so, no plant sample collections have been done in maize?

Reply: See line 163-167

Lines138-139: here is reported soil collections in maize, but no plant samples have been collected? Is there a mistake? Please specify. In addition why soil samples in wheat have been collected between rows and in maize within rows? These aspects should be clarified.

Reply: The fiirst sentence alarify the collection of plant samples. Because the distance between two plants of wheat and maize is different, we collect the soil of root concentrated as far as possible.

Line 166: this section should be better described. I am not sure regarding the experimental design do you adopted for the statistical analysis. How the growin season was treated? These aspects should be clear for the readers.

Reply: Modified, see line 198-199

RESULTS:

Figure 1: it is too small and it is difficult to read the letters, I suggest to report the figures vertically and increase the size. I am not sure (but small size) the unit of measure is correct, please check. What means the vertical bars? This should be clarified in the figure caption.

Reply: Modified, see new version, line 205.

Line 178: 30 days after fertilization means that it corresponds with “seedling stage”? if so please specify.

Reply: Modified, see line 221-222

Lines177-185: the two figures should be better discussed. Now there is confusion. I suggest to rewrite starting with the description of the figure 1a and then figure 1b. In this way it is more clear the results related to the soil NH4 content during the main crop stages.

Reply: Modified, see line 209-217

Figure 2. It is very difficult to see the differences among the treatments, the figures should be improved. I suppose that these measurements are related to the maize crop, but it is not clear for me. In addition, in the figure is reported horizontal rows that should be explained in the figure caption.

Reply: Modified, changed the new picture.

Line 202: please avoid the term “For example”, here you are reporting the findings of your study, this need to be clear and concise.

Reply: Modified, deleted “for example, see line 236

Figure 3. The figure format should be improved, it is very small to be read.

Reply: The treatments and growing period are a lilttle long, it is not easy to lay if the figures are enlarged.

Figure 4. The same consideration of Fig 2. 30, 240 and 330 days from what? This is not clear and could make confusion for a potential reader.

Reply: Modified, changed the new picture.

Table 1. I suggest removing doubled information. One way could be to report the harvest index of wheat and maize instead of biomass of each crop (it is the sum of the previous data), while in the total yield, I suggest removing grain and straw and leave the total biomass. The same could be done in Table 2.

Reply: Modified, see line 342

In all tables remove the note and add the information in the caption. Reply: Modified

DISCUSSION:

The application of organic N fertilizers contributes to reduce the N mineralization and thus N losses. Although it is not considered as experimental treatment in this study, I suggest adding some sentences related to the importance of organic fertilizers could play in the N cycling.

Reply: Because all seven treatments are applied with same organic fertilizer treatment, we thinlk it is not necessary to discuss this topic.

I suggest also to report more implication that the adoption of N inhibitors could have in the crop management and the benefits obtained from this experiment. Moreover, how these findings could be applied in the next study?

Reply: Modified, add some words, see line 539-540

This manuscript is a resubmission of an earlier submission. The following is a list of the peer review reports and author responses from that submission.

Round 1

Reviewer 1 Report

Potential the manuscript is interesting, but the cultivation in open field of each crop for only one year is not sufficient to draw good results.

Introduction is too short. Authors need to improve it.

Main Max and min temperatures should be added as well as all, the agromomic managment of both the assessed crops.

Reviewer 2 Report

Dear Authors,

Thank you for giving me the possibility to review the manuscript n. agronomy-1617805 entitled “Yield effect and space-time changes of soil nitrogen under different urease/nitrate inhibitor combined in wheat/maize rotation system”.

GENERAL CONSIDERATIONS

The manuscript goal is strongly related to one of the main aspects of the modern agriculture i.e. the tentative to reduce the amount of external inputs (fertilizers) by improving their use efficiency. The topic fits with the journal aims. However, one of the main critical aspects is related to the experimental period. In the manuscript it is reported “Experiments were carried out at a seed-multiplication farm (E 116°89′-116°90′, N 79 35°29′-35°30′) located in Shiqiang Town, Zoucheng City, Shandong Province from October 2018 to October 2020” but nothing is reported regarding the growing cycle of each crop. For an important journal with international share the agronomical trials should be repeated at least two times, specifically this means that the results should report 2 sequences of wheat-maize. This should be reported clearly. In addition, the statistical analysis should be improved to have a clear idea of data management.

ABSTRACT: The abstract is too long and should be shortened to 200 words according with the Instructions for Authors. In addition, the headings (background, Methods, Results and Conclusions) should be removed.

KEY WORDS: Most of the key words are already reported in the title, therefore, they do not add any additional information to the potential readers. I suggest improving the manuscript with other key words.

INTRODUCTION:

Lines 43 – 45 -> These short sentences could be grouped in one sentence describing better the need of nitrogen fertilizers for improving/maintaining crop yields in the agricultural sector. In addition, the connection with the N fertilizer use in China should be gradually introduced by describing the high use of nitrogen fertilizers worldwide.

Line 60 -> Please check the temperature 10 °C is reported twice.

Line 70 -> In my opinion, it is better to write a manuscript in third person avoiding objective considerations or sentences. Please convert the sentence in third person.

Lines 72 – 75 -> This sentence is a part of Materials and Methods, please remove it.

Lines 75 – 76 -> This sentence is a part of discussion, Please remove it.

The introduction section is not completely developed. It should be improved adding the clear effect of nitrogen fertilization on crop yield and overall by discussing the current knowledge on the space-time changes of soil nitrogen related to the different dynamics that could be interested by the N fertilizations. In addition, here it is missing the hypothesis of the study (what do the authors assume with the research activity in this study?). Moreover, the introduction should clearly report the objectives of the study.

MATERIALS AND METHODS

Lines 84 – 87 and in the whole manuscript -> unit of measure should be reported in the following way: mg kg-1, I suggest to convert all unit of measure

Line 104 -> The described treatments are seven and not height please clarify this point.

Lines 114 – 116 -> The authors should explain why they adopted the described amount of N, P2O5 and K2O (common practices adopted by the local farmers?). In addition, it is not clear if the same amount was applied in wheat and maize, or it is the total rate. Please this section is extremely important to have a clear vision of the experiment. I suggest rewriting in deep to avoid misunderstanding for the potential readers. It is also important to state if the fertilizers were applied in one time or splitted in separate times.

Lines 154 – 160 -> Please add the references related to the NUE index adopted in this manuscript.

Lines 163 – 165 -> The statistical approach should be improved. How was the year considered? How was the average value separated? Duncan, Fisher test? I suggest rewriting this section with more detailed information.

RESULTS

The results analyze also the N form (nitrate and ammonium) in the soil at different soil depth. It could be good idea to report also the data related to the meteorological conditions, especially rainfall and/or evapotranspiration, during the experimental period (there is some statement in the discussion, but specific data could improve the quality of the discussion). It could be useful for explain leaching dynamics.

Lines 168 – 179 -> This pard of the manuscript is descriptive and it is not completely related to the results observed in figures. I suggest moving in the discussion section.

FIGURE 1 -> the figure caption should report also that the measurements are performed at different level of wheat and maize growing stage.

TABLE 2 -> it is missing the statement regarding the meaning of letters. Please add.

Based on the above consideration, I appreciate the work done by the authors, however I regret to inform that at this stage the manuscript could not be accepted for publication in the journal. I suggest to improve it following the suggestion and resubmit for further consideration.

Reviewer 3 Report

The article presented for review is interesting and brings a lot of information to science, especially for to plant production and environmental protection.

However, I believe that the research should be treated as preliminary (one year of research) and continued. One cannot draw firm conclusions on the basis of one-year analyzes, especially in such a changeable environment as soil is.